# Object-Oriented Dynamics Predictor

**Guangxiang Zhu, Zhiao Huang, and Chongjie Zhang**
Institute for Interdisciplinary Information Sciences
Tsinghua University, Beijing, China
`guangxiangzhu@outlook.com,hza14@mails.tsinghua.edu.cn,chongjie@tsinghua.edu.cn`

## Abstract

Generalization has been one of the major challenges for learning dynamics models in model-based reinforcement learning. However, previous work on action-conditioned dynamics prediction focuses on learning the pixel-level motion and thus does not generalize well to novel environments with different object layouts. In this paper, we present a novel object-oriented framework, called *object-oriented dynamics predictor* (OODP), which decomposes the environment into objects and predicts the dynamics of objects conditioned on both actions and object-to-object relations. It is an end-to-end neural network and can be trained in an unsupervised manner. To enable the generalization ability of dynamics learning, we design a novel CNN-based relation mechanism that is class-specific (rather than object-specific) and exploits the locality principle. Empirical results show that OODP significantly outperforms previous methods in terms of generalization over novel environments with various object layouts. OODP is able to learn from very few environments and accurately predict dynamics in a large number of unseen environments. In addition, OODP learns semantically and visually interpretable dynamics models.

## 1 Introduction

Recently, model-free deep reinforcement learning (DRL) has been extensively studied for automatically learning representations and decisions from visual observations to actions. Although such approach is able to achieve human-level control in games [1, 2, 3, 4, 5], it is not efficient enough and cannot generalize across different tasks. To improve sample-efficiency and enable generalization for tasks with different goals, model-based DRL approaches (e.g., [6, 7, 8, 9]) are extensively studied.

One of the core problems of model-based DRL is to learn a dynamics model through interacting with environments. Existing work on learning action-conditioned dynamics models [10, 11, 12] has achieved significant progress, which learns dynamics models and achieves remarkable performance in training environments [10, 11], and further makes a step towards generalization over object appearances [12]. However, these models focus on learning pixel-level motions and thus their learned dynamics models do not generalize well to novel environments with different object layouts. Cognitive studies show that objects are the basic units of decomposing and understanding the world through natural human senses [13, 14, 15, 16], which indicates object-based models are useful for generalization [17, 18, 19].

In this paper, we develop a novel object-oriented framework, called *object-oriented dynamics predictor* (OODP). It is an end-to-end neural network and can be trained in an unsupervised manner. It follows an object-oriented paradigm, which decomposes the environment into objects and learns the object-level dynamic effects of actions conditioned on object-to-object relations. To enable the generalization ability of OODP's dynamics learning, we design a novel CNN-based relation mechanism that is class-specific (rather than object-specific) and exploits the locality principle. This mechanism induces neural networks to distinguish objects based on their appearances, as well as their effects on an agent's dynamics.

Empirical results show that OODP significantly outperforms previous methods in terms of generalization over novel environments with different object layouts. OODP is able to learn from very few environments and accurately predict the dynamics of objects in a large number of unseen environments. In addition, OODP learns semantically and visually interpretable dynamics models, and demonstrates robustness to some changes of object appearance.

## 2 Related Work

**Action-conditioned dynamics learning approaches** have been proposed for learning the dynamic effects of an agent's actions from raw visual observations and achieves remarkable performance in training environments [10, 11]. CDNA [12] further makes a step towards generalization over object appearances, and demonstrates its usage for model-based DRL [9]. However, these approaches focus on learning pixel-level motions and do not explicitly take object-to-object relations into consideration. As a result, they cannot generalize well across novel environments with different object layouts. An effective relation mechanism can encourage neural networks to focus attentions on the moving objects whose dynamics needs to be predicted, as well as the static objects (e.g., walls and ladders) that have an effect on the moving objects. The lack of such a mechanism also accounts for the fact that the composing masks in CDNA [12] are insensitive to static objects.

**Relation-based nets** have shown remarkable effectiveness to learn relations for physical reasoning [19, 20, 21, 22, 23, 24]. However, they are not designed for action-conditioned dynamics learning. In addition, they have either manually encoded object representations [19, 20] or not demonstrated generalization across unseen environments with novel object layouts [20, 21, 22, 23, 24]. In this paper, we design a novel CNN-based relation mechanism. First, this mechanism formulates a spatial distribution of a class of objects as a class-specific object mask, instead of representing an individual object by a vector, which allows relation nets to handle a vast number of object samples and distinguish objects by their specific dynamic properties. Second, we use neighborhood cropping and CNNs to exploit the locality principle of object interactions that commonly exists in the real world.

**Object-oriented reinforcement learning** has been extensively studied, whose paradigm is that the learning is based on object representations and the effects of actions are conditioned on object-to-object relations. For example, *relational MDPs* [17] and *Object-Oriented MDPs* [18] are proposed for efficient model-based planning or learning, which supports strong generalization. Cobo et al. [25] develop a model-free object-oriented learning algorithm to speed up classic Q-learning with compact state representations. However, these models require explicit encoding of the object representations [17, 18, 25] and relations [17, 18]. In contrast, our work aims at automatically learning object representations and object-to-object relations to support model-based RL. While approaches from object localization [26] or disentangled representations learning [27, 28] have been proposed for identifying objects, unlike our model, they cannot perform action-conditioned relational reasoning to enable generalizable dynamics learning.

## 3 Object-Oriented Dynamics Predictor

To enable the generalization ability over object layouts for dynamics learning, we develop a novel unsupervised end-to-end neural network framework, called *Object-Oriented Dynamics Predictor* (OODP). This framework takes the video frames and agents' actions as input and learns the dynamics of objects conditioned on actions and object-to-object relations. Figure 1 illustrates the framework of OODP that consists of three main components: Object Detector, Dynamics Net, and Background Extractor. In this framework, the input image is fed into two data streams: dynamics inference and background extraction. In the dynamics inference stream, Object Detector decomposes the input image into dynamic objects (e.g., the agent) and static objects (e.g., walls and ladders). Then, Dynamic Net learns the motions of dynamic objects based on both their relations with other objects and the actions of an agent (e.g., the agent's moving upward by actions *up* when touching a ladder). Once the motions are learned, we can apply these transformations to the dynamic objects via Spatial Transformer Network (STN) [29]. In the background extraction stream, Background Extractor extracts the background of the input image, which is not changing over time. Finally, the extracted background will be combined with the transformed dynamic objects to generate the prediction of the

next frame. OODP assumes the environment is Markovian and deterministic, so it predicts the next frame based on the current frame and action.

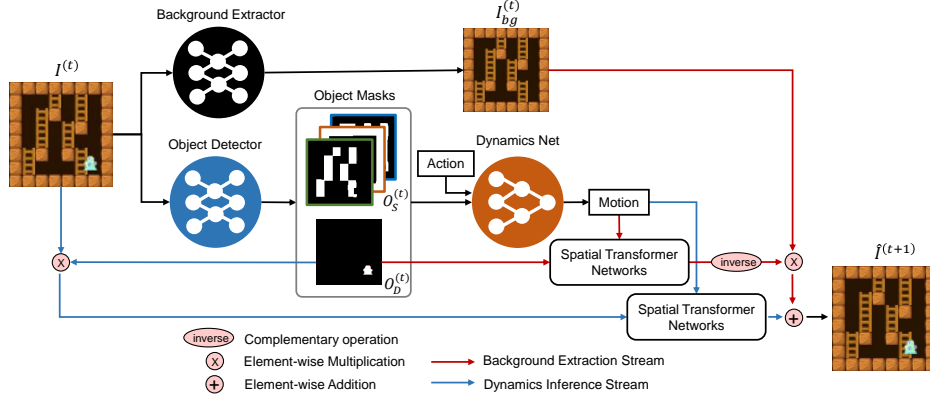

Figure 1: Overall framework of OODP.

OODP follows an object-oriented dynamics learning paradigm. It uses object masks to bridge object perception (Object Detector) with dynamics learning (Dynamics Net) and to form a tensor bottleneck, which only allows the object-level information to flow through. Each object mask has its own dynamics learner, which forces Object Detector to act as a detector for the object of interest and also as a classifier for distinguishing which kind of object has a specific effect on dynamics. In addition, we add an entropy loss to reduce the uncertainty of object masks, thus encouraging attention on key parts and learning invariance to irrelevances.

In the rest of this section, we will describe in detail the design of each main component of OODP and their connections.

## 3.1 Object Detector

Object Detector decomposes the input image into a set of objects. To simplify our model, we group the objects (denoted as $O$) into static and dynamic objects (denoted as $S$ and $D$) so that we only need to focus on predicting the motions of the dynamic objects. An object $O_i$ is represented by a mask $M_{O_i} \in [0, 1]^{H \times W}$, where $H$ and $W$ denote the height and width of the input image $I \in \mathbb{R}^{H \times W \times 3}$, and the entry $M_{O_i}(u, v)$ indicates the probability that the pixel $I(u, v)$ belongs to the $i$-th object. The same class of static objects has the same effects on the motions of dynamic objects, so we use one mask $M_{S_i}$ ($1 \le i \le n_S$, where $n_S$ denotes the class number of static objects) to represent each class of static objects. As different dynamic objects may have different motions, we use one mask $M_{D_j}$ ($1 \le j \le n_D$, where $n_D$ denotes the individual number of dynamic objects) to represent each individual dynamic object. Note that OODP does not require the actual number of objects in an environment, but needs to set a maximum number. When they do not match, some learned object masks may be redundant, which does not affect the accuracy of prediction (more details can be found in Supplementary Material).

Figure 2 depicts the architecture of Object Detector. As shown in the figure, the pixels of the input image go through different CNNs to obtain different object masks. There are totally $n_O$ CNNs owning the same architecture but not sharing weights. The architecture details of these CNNs can be found in Supplementary Material. Then, the output layers of these CNNs are concatenated with each other across channels and a pixel-wise softmax is applied on the concatenated feature map $F \in \mathbb{R}^{H \times W \times n_O}$. Let $f(u, v, c)$ denote the value at position $(u, v)$ in the $c$-th channel of $F$. The entry $M_{O_c}(u, v)$ of the $c$-th object mask which represents the probability that the pixel $I(u, v)$ of the input image belongs to the $c$-th object $O_c$, can be calculated as,

$$M_{O_c}(u, v) = p\left(O_c | I(u, v)\right) = \frac{e^{f(u, v, c)}}{\sum_i^{n_O} e^{f(u, v, i)}}.$$

To reduce the uncertainty of the affiliation of each pixel $I(u, v)$ and encourage the object masks to obtain more discrete attention distributions, we introduce a pixel-wise entropy loss to limit the

entropy of the object masks, which is defined as,

$$\mathcal{L}_{\text{entropy}} = \sum_{u,v} \sum_{c}^{n_O} -p\left(O_c | I(u,v)\right) \log p\left(O_c | I(u,v)\right).$$

## 3.2 Dynamics Net

Dynamics Net aims at learning the motion of each dynamic object conditioned on actions and object-to-object relations. Its architecture is illustrated in Figure 3. In order to improve the computational efficiency and generalization ability, the Dynamics Net architecture incorporates a tailor module to exploit the locality principle and employs CNNs to learn the effects of object-to-object relations on the motions of dynamic objects. As the locality principle commonly exists in object-to-object relations in the real world, the tailor module enables the inference on the dynamics of objects focusing on the relations with neighbour objects. Specifically, it crops a "horizon" window of size $w$ from the object masks centered on the position of the dynamic object whose motion is being predicted, where $w$ indicates the maximum effective range of object-to-object relations. The tailored local objects are then fed into the successive network layers to reason their effects. Unlike most previous work which uses fully connected networks for identifying relations [19, 20, 21, 22, 23], our dynamics inference employs CNNs. This is because CNNs provide a mechanism of strongly responding to spatially local patterns, and the multiple receptive fields in CNNs are capable of dealing with complex spatial relations expressed in distribution masks.

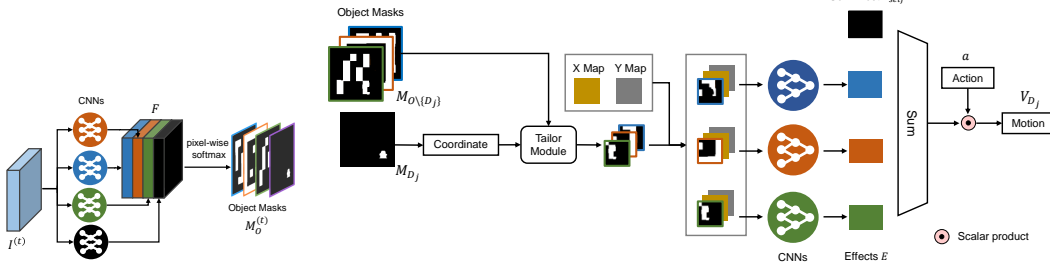

Figure 2: Architectures of Object Detector.

Figure 3: Architecture of Dynamics Net. We illustrate predicting the motion of $M_{D_j}$ as an example. $O\backslash\{D_j\}$ denotes $O$ excluding $D_j$.

To demonstrate the detailed pipeline in Dynamics Net (Figure 3), we take as an example the computation of the predicted motion of the $j$-th dynamic object $D_j$. First, we describe the cropping process of the tailor module, which crops the object masks near to the dynamic object $D_j$.

For each dynamic object $D_j$, its position $(\bar{u}_{D_j}, \bar{v}_{D_j})$ is defined as the expected location of its object mask $M_{D_j}$, where $\bar{u}_{D_j} = \left(\sum_u^H \sum_v^W M_{D_j}(u,v)\right)^{-1} \sum_u^H \sum_v^W u \cdot M_{D_j}(u,v)$, $\bar{v}_{D_j} = \left(\sum_u^H \sum_v^W M_{D_j}(u,v)\right)^{-1} \sum_u^H \sum_v^W v \cdot M_{D_j}(u,v)$. The "horizon" window of size $w$ centered on $(\bar{u}_{D_j}, \bar{v}_{D_j})$ is written as $B_w = \{(u,v) : \bar{u}_{D_j} - w/2 \le u \le \bar{u}_{D_j} + w/2, \bar{v}_{D_j} - w/2 \le v \le \bar{v}_{D_j} + w/2\}$. The tailor module receives $B_w$ and performs cropping on other object masks excluding $M_{D_j}$, that is, $M_{O\backslash\{D_j\}}$. This cropping process can be realized by bilinear sampling [29], which is a sub-differentiable sampling approach. Taking cropping $M_{O_1}$ as an example, the pairwise transformation of sampling grid is $(u_2, v_2) = (u_1 + \bar{u}_{D_j} - w/2, v_1 + \bar{v}_{D_j} - w/2)$, where $(u_1, v_1)$ are coordinates of the cropped object mask $C_{O_1}$ and $(u_2, v_2) \in B_w$ are coordinates of the original object mask $M_{O_1}$. Then applying bilinear sampling kernel on this grid can compute the cropped object mask $C_{O_1}$,

$$C_{O_1}(u_1, v_1) = \sum_u^H \sum_v^W \left( M_{O_1}(u,v) \cdot \max(0, 1 - |u_2 - u|) \max(0, 1 - |v_2 - v|) \right), \quad (1)$$

Note that the gradients wrt $(\bar{u}_{D_j}, \bar{v}_{D_j})$ are frozen to force the cropping module focus on what to crop rather than where to crop, which is different to the vanilla bilinear sampling [29].

Then, each cropped object mask $C_{O_i}$ is concatenated with the constant x-coordinate and y-coordinate meshgrid map, which makes networks more sensitive to the spatial information. The concatenated

map is fed into its own CNNs to learn the effect of $i$-th object on $j$-th dynamic object $E(O_i, D_j) \in \mathbb{R}^{2 \times n_a}$, where $n_a$ represents the number of actions. There are totally $(n_O - 1) \times n_D$ CNNs for $(n_O - 1) \times n_D$ pairs of objects. Different CNNs working for different objects forces the object mask to act as a classifier for distinguishing each object with the specific dynamics. To predict the motion vector $V_{D_j}^{(t)} \in \mathbb{R}^2$ for dynamic object $D_j$, all the object effects and a self effect $E_{\text{self}}(D_j) \in \mathbb{R}^{2 \times n_a}$ representing the natural bias are summed and multiplied by the one-hot coding of action $a^{(t)} \in \{0, 1\}^{n_a}$, that is,

$$V_{D_j}^{(t)} = \Big( \big( \sum_{O_i \in O \setminus \{D_j\}} E(O_i, D_j) \big) + E_{\text{self}}(D_j) \Big) \cdot a^{(t)}.$$

In addition to the conventional prediction error $\mathcal{L}_{\text{prediction}}$ (described in Section 3.4), here we introduce a regression loss to guide the optimization of $M_O^{(t)}$ and $V_D^{(t)}$, which serves as a highway to provide early feedback before reconstructing images. This regression loss is defined as follows,

$$\mathcal{L}_{\text{highway}} = \sum_j^{n_D} \left\| (\bar{u}_{D_j}, \bar{v}_{D_j})^{(t)} + V_{D_j}^{(t)} - (\bar{u}_{D_j}, \bar{v}_{D_j})^{(t+1)} \right\|_2^2.$$

### 3.3 Background Extractor

To extract time-invariant background, we construct a Background Extractor with neural networks. The Background Extractor takes the form of a traditional encoder-decoder structure. The encoder alternates convolutions [30] and Rectified Linear Units (ReLUs) [31] followed by two fully-connected layers, while the decoder alternates deconvolutions [32] and ReLUs. To avoid losing information in pooling layers, we replace all the pooling layers by convolutional layers with increased stride [33, 34]. Further details of the architecture of Background Extractor can be found in Supplementary Material.

Background Extractor takes the current frame $I^{(t)} \in \mathbb{R}^{H \times W \times 3}$ as input and produces the background image $I_{\text{bg}}^{(t)} \in \mathbb{R}^{H \times W \times 3}$, whose pixels remain unchanged over times. We use $\mathcal{L}_{\text{background}}$ here to force the network to satisfy such a property of time invariance, given by $\mathcal{L}_{\text{background}} = \left\| I_{\text{bg}}^{(t+1)} - I_{\text{bg}}^{(t)} \right\|_2^2$.

### 3.4 Merging Streams

Finally, two streams of pixels are merged to produce the prediction of the next frame. One stream carries the pixels of dynamic objects which can be obtained by transforming the masked pixels of dynamic objects from the current frame. The other stream provides the rest pixels generated by Background Extractor.

Spatial Transformer Network (STN) [29] provides neural networks capability of spatial transformation. In the first stream, a STN accepts the learned motion vectors $V$ and performs spatial transforms (denoted by $\text{STN}(*, V)$) on dynamic pixels. The bilinear sampling (similar as Equation 1) is also used in STN to compute the transformed pixels in a sub-differentiable manner. The difference is that, we allow the gradients of loss backpropagate to the object masks as well as the motion vectors. Then, we obtain the pixel frame $\hat{I}_1^{(t+1)}$ of dynamic objects in the next frame $\hat{I}^{(t+1)}$, that is, $\hat{I}_1^{(t+1)} = \sum_j^{n_D} \text{STN}(M_{D_j}^{(t)} \cdot I^{(t)}, V_{D_j}^{(t)})$, where $\cdot$ denotes element-wise multiplication. The other stream aims at computing the rest pixels $\hat{I}_2^{(t+1)}$ of $\hat{I}^{(t+1)}$, that is, $\hat{I}_2^{(t+1)} = \big( 1 - \sum_j^{n_D} \text{STN}(M_{D_j}^{(t)}, V_{D_j}^{(t)}) \big) \cdot I_{\text{bg}}^{(t)}$. Thus, the output end of OODP, $\hat{I}^{(t+1)}$, is calculated by,

$$\hat{I}^{(t+1)} = \hat{I}_1^{(t+1)} + \hat{I}_2^{(t+1)} = \sum_j^{n_D} \text{STN}(M_{D_j}^{(t)} \cdot I^{(t)}, V_{D_j}^{(t)}) + \big( 1 - \sum_j^{n_D} \text{STN}(M_{D_j}^{(t)}, V_{D_j}^{(t)}) \big) \cdot I_{\text{bg}}^{(t)}.$$

We use $l_2$ pixel loss to restrain image prediction error, which is given by, $\mathcal{L}_{\text{prediction}} = \left\| \hat{I}^{(t+1)} - I^{(t+1)} \right\|_2^2$. We also add a similar $l_2$ pixel loss for reconstructing the current image, that is,

$$\mathcal{L}_{\text{reconstruction}} = \left\| \sum_j^{n_D} M_{D_j}^{(t)} \cdot I^{(t)} + \big( 1 - \sum_j^{n_D} M_{D_j}^{(t)} \big) \cdot I_{\text{bg}}^{(t)} - I^{(t)} \right\|_2^2.$$

In addition, we add a loss $\mathcal{L}_{\text{consistency}}$ to link the visual perception and dynamics prediction of the objects, which enables learning object by integrating vision and interaction,

$$\mathcal{L}_{\text{consistency}} = \sum_{j}^{n_D} \left\| M_{D_j}^{(t+1)} - \text{STN}\big(M_{D_j}^{(t)}, V_{D_j}^{(t)}\big) \right\|_2^2 .$$

### 3.5 Training Procedure

OODP is trained by the following loss given by combining the previous losses with different weights,

$$\mathcal{L}_{\text{total}} = \mathcal{L}_{\text{highway}} + \lambda_p \mathcal{L}_{\text{prediction}} + \lambda_e \mathcal{L}_{\text{entropy}} + \lambda_r \mathcal{L}_{\text{reconstruction}} + \lambda_c \mathcal{L}_{\text{consistency}} + \lambda_{bg} \mathcal{L}_{\text{background}}$$

Considering that signals derived from foreground detection can benefit Object Detector to produce more accurate object masks, we use the simplest unsupervised foreground detection approach [35] to calculate a rough proposal dynamic region and then add a $l_2$ loss to encourage the dynamic object masks to concentrate more attentions on this region, that is,

$$\mathcal{L}_{\text{proposal}} = \big\| \big(\sum_{j}^{n_D} M_{D_j}^{(t)}\big) - M_{\text{proposal}}^{(t)} \big\|_2^2, \tag{2}$$

where $M_{\text{proposal}}^{(t)}$ represents the proposal dynamic region. This additional loss can facilitate the training and make the learning process more stable and robust.

## 4 Experiments

### 4.1 Experiment Setting

We evaluate our model on *Monster Kong* from the Pygame Learning Environment [36], which offers various scenes for testing generalization abilities across object layouts (e.g., different number and spatial arrangement of objects). Across different scenes, the same underlying physics engine that simulates the real-world dynamics mechanism is shared. For example, in each scene, an agent can move upward using a ladder, it will be stuck when hitting the walls, and it will fall in free space. The agent explores various environments with a random policy over actions including *up*, *down*, *left*, *right*, and *noop* and its gained experiences are used for learning dynamics. The code of OODP is available online at `https://github.com/mig-zh/OODP`.

To evaluate the generalization ability, we compare our model with state-of-the-art action-conditioned dynamics learning approaches, AC Model [10], and CDNA [12]. We evaluate all models in $k$-to-$m$ zero-shot generalization problems (Figure 4), where they learn dynamics with $k$ different training environments and are then evaluated in $m$ different unseen testing environments with different object layouts. In this paper, we use $m = 10$ and $k = 1, 2, 3, 4$, and 5, respectively. The smaller the value $k$, the more difficult the generalization problem. In this setting, truly achieving generalization to new scenes requires learners' full understanding of the object-level abstraction, object relationships and dynamics mechanism behind the images, which is quite different from the conventional video prediction task and crucially challenging for the existing learning models. In addition, we will investigate whether OODP can learn semantically and visually interpretable knowledge and is robustness to some changes of object appearance.

### 4.2 Zero-shot Generalization of Learned Models

To demonstrate the generalization ability, we evaluate the prediction accuracy of the learned dynamics model in unseen environments with novel object layouts without re-training. Table 1 shows the performance of all models on predicting the dynamics of the agent, where $n$-error accuracy is defined as the proportion that the difference between the predicted and ground-true agent locations is less than $n$ pixel ($n = 0, 1, 2$).

From Table 1, we can see our model significantly outperforms the previous methods under all circumstances. This demonstrates our object-oriented approach is beneficial for the generalization over object layouts. As expected, as the number of training environments increases, the learned object dynamics can generalize more easily over new environments and thus the accuracy of dynamics

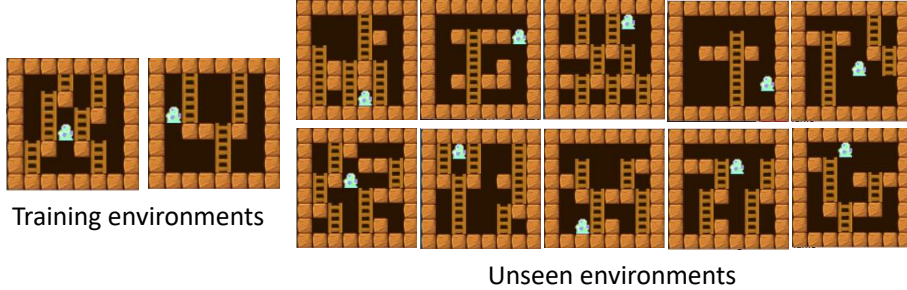

Training environments

Unseen environments

Figure 4: An example of 2-to-10 zero-shot generalization problem.

|  | Models | Training environments | | | | | Unseen environments | | | | |
|---|---|---|---|---|---|---|---|---|---|---|---|
|  |  | 1-10 | 2-10 | 3-10 | 4-10 | 5-10 | 1-10 | 2-10 | 3-10 | 4-10 | 5-10 |
| 0-error accuracy | OODP+p | 0.90 | 0.94 | 0.92 | 0.93 | 0.93 | **0.32** | **0.78** | 0.73 | 0.79 | 0.82 |
|  | OODP-p | 0.96 | 0.98 | 0.98 | 0.98 | 0.97 | 0.22 | **0.78** | **0.86** | **0.90** | **0.95** |
|  | AC Model | 0.99 | 0.99 | 0.99 | 0.99 | 0.99 | 0.01 | 0.17 | 0.22 | 0.44 | 0.70 |
|  | CDNA | 0.20 | 0.13 | 0.14 | 0.19 | 0.17 | 0.33 | 0.18 | 0.20 | 0.25 | 0.19 |
| 1-error accuracy | OODP+p | 0.98 | 0.98 | 0.99 | 0.98 | 0.98 | **0.71** | 0.90 | 0.90 | 0.94 | 0.95 |
|  | OODP-p | 0.98 | 0.98 | 0.99 | 0.99 | 0.99 | 0.61 | **0.91** | **0.94** | **0.96** | **0.97** |
|  | AC Model | 0.99 | 0.99 | 0.99 | 0.99 | 0.99 | 0.01 | 0.31 | 0.31 | 0.57 | 0.77 |
|  | CDNA | 0.30 | 0.29 | 0.30 | 0.33 | 0.30 | 0.53 | 0.49 | 0.47 | 0.52 | 0.55 |
| 2-error accuracy | OODP+p | 0.99 | 0.99 | 0.99 | 0.99 | 0.99 | **0.87** | **0.96** | 0.96 | **0.98** | **0.99** |
|  | OODP-p | 0.99 | 0.99 | 0.99 | 0.99 | 0.99 | 0.82 | 0.94 | **0.97** | **0.98** | 0.98 |
|  | AC Model | 0.99 | 0.99 | 0.99 | 0.99 | 0.99 | 0.02 | 0.37 | 0.34 | 0.64 | 0.80 |
|  | CDNA | 0.36 | 0.44 | 0.47 | 0.45 | 0.45 | 0.56 | 0.55 | 0.56 | 0.62 | 0.62 |

Table 1: Accuracy of the dynamics prediction. $k$-$m$ means the $k$-to-$m$ zero-shot generalization problem. Here, we use OODP+p and OODP-p to distinguish OODP with or without the proposal loss (Equation 2).

prediction tends to grow. OODP achieves reasonable performance with 0.86 0-error accuracy only trained in 3 environments, while the other methods fail to get satisfactory scores (about 0.2). We observe that the AC Model achieves extremely high accuracy in training environments but cannot make accurate predictions in novel environments, which implies it overfits the training environments severely. This is partly because the AC Model only performs video prediction at the pixel level and learns few object-level knowledge. Though CDNA includes object concepts in their model, it still performs pixel-level motion prediction and does not consider object-to-object relations. As a result, CDNA also fails to achieve accurate predictions in unseen environments with novel object layouts (As the tuning of hyper parameters does not improve the prediction performance, we use the default settings here). In addition, we observe that the performance of OODP-p is slightly higher than OODP+p because the region proposals used for the initial guidance of optimization sometimes may introduce some noise. Nevertheless, using proposals can make the learning process more stable.

## 4.3 Interpretability of Learned Knowledge

Interpretable deep learning has always been a significant but vitally hard topic [37, 38, 39]. Unlike previous video prediction frameworks [10, 12, 40, 41, 42, 43, 23], most of which use neural networks with uninterpretable hidden layers, our model has informative and meaningful intermediate layers containing the object-level representations and dynamics.

To interpret the intermediate representations learned by OODP, we illustrate its object masks in unseen environments, as shown in Figure 5. Intriguingly, the learned dynamic object masks accurately capture the moving agents, and the static masks successfully detect the ladders, walls and free space that lead to different action conditioned dynamics of the agent. Each object mask includes one class of objects, which implies that the common characteristics of this class are learned and the knowledge that links

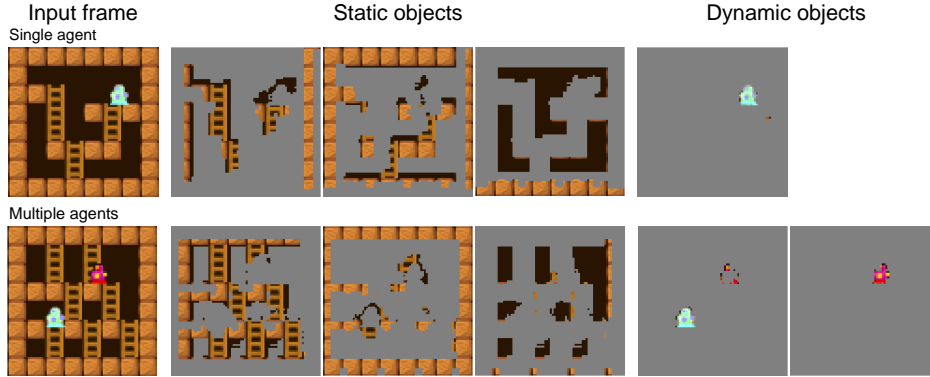

Figure 5: Visualization of the masked images in unseen environments with single dynamic object (top) and multiple dynamic objects (down). To demonstrate the learned attentions of object masks, the raw input images are multiplied by binarized object masks.

visual features and dynamics properties is gained. While the learned masks are not as fine as those derived from the supervised image segmentation, they clearly demonstrate visually interpretable representations in the domain of unsupervised dynamics learning.

To interpret the learned object dynamics behind frame prediction, we evaluate the root-mean-square errors (RMSEs) between the predicted and ground-truth motions. Table 2 shows the RMSEs averaged over 10000 samples. From Table 2, we can observe that motions predicted by OODP are very accurate, with the RMSE close or below one pixel, in both training and unseen environments. Such a small error is visually indistinguishable since it is less than the resolution of the input video frame (1 pixel). As expected, as the number of training environments increases, this prediction error rapidly descends. Further, we also provide some intuitive prediction examples (see Supplementary Material) and a video (https://goo.gl/BTL2wH) for better perceptual understanding of the prediction performance.

| | Models | Number of training envs | | | | |
| | | 1 | 2 | 3 | 4 | 5 |
|---|---|---|---|---|---|---|
| Training envs | OODP+p | 0.28 | 0.24 | 0.23 | 0.23 | 0.23 |
| | OODP-p | 0.18 | 0.17 | 0.19 | 0.14 | 0.15 |
| Unseen envs | OODP+p | 1.04 | 0.52 | 0.51 | 0.43 | 0.40 |
| | OODP-p | 1.09 | 0.53 | 0.38 | 0.35 | 0.29 |

Table 2: RMSEs between predicted and ground-truth motions. The unit of measure is pixel.

| | Object appearance | | | | | | |
| | S0 | S1 | S2 | S3 | S4 | S5 | S6 |
|---|---|---|---|---|---|---|---|
| Acc | 0.94 | 0.92 | 0.94 | 0.94 | 0.92 | 0.88 | 0.93 |
| RMSE | 0.29 | 0.35 | 0.31 | 0.28 | 0.31 | 0.40 | 0.30 |

Table 3: The performance (accuracy and RMSE in 5-to-10 zero-shot generalization problem) of OODP in novel environments with different object layouts and appearances.

These interpretable intermediates demystify why OODP is able to generalize across novel environments with various object layouts. While the visual perceptions of novel environments are quite different, the underlying physical mechanism based on object relations keeps invariant. As shown in Figure 5, OODP learns to decompose a novel scene into understandable objects and thus can reuse object-level knowledge (features and relations) acquired from training environments to predict the effects of actions.

## 4.4 Robustness to Changes of Object Appearance

To demonstrate the robustness to object appearances, we evaluate the generalization performance of OODP in testing environments including objects with appearance differences from those in training environments, as shown in Figure 6. As shown in Table 3, OODP provides still high prediction performance in all these testing environments, which indicates that it can still generalize to novel object layouts even when the object appearances have some differences. This robustness is partly because the Object Detector in OODP employs CNNs that are capable of learning essential patterns

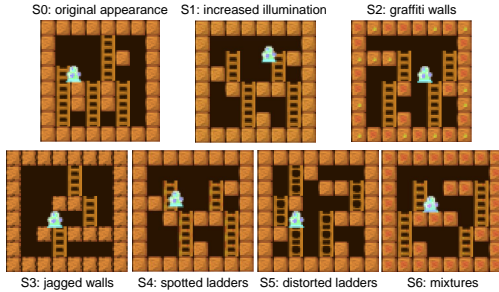

S0: original appearance  S1: increased illumination  S2: graffiti walls

S3: jagged walls  S4: spotted ladders  S5: distorted ladders  S6: mixtures

Figure 6: Illustration of the configurations of the novel testing environments. Compared to the training environments, the testing ones have different object layouts (S0-S6), and their objects have some appearance differences (S1-S6).

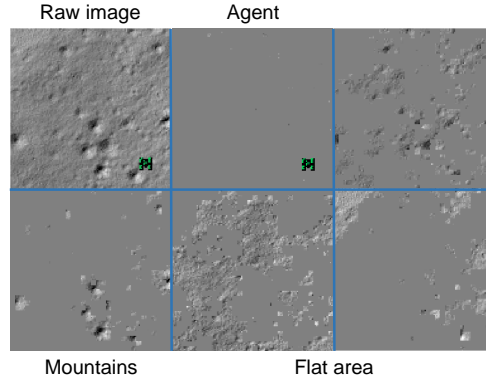

Raw image  Agent

Mountains  Flat area

Figure 7: Visualization of the learned masks in unseen environments in Mars domain.

over appearances. Furthermore, we provide the learned masks (see Supplementary Material) and a video (`https://goo.gl/ovupdn`) to show our results on S2 environments.

### 4.5 Performance on Natural Image Input

To test the performance for the natural image input, we also evaluate our model in the Mars Rover Navigation domain introduced by Tamar et al. [44]. The Mars landscape images are natural images token from NASA. A Mars rover random explores in the Martian surface and it will be stuck if there are mountains whose elevation angles are equal or greater than 5 degrees. We run our model on 5-to-10 zero-shot generalization problem and compare it with other approaches. As shown in Figure 7 and Table 4, in unseen environments, our learned object masks successfully capture the key objects and our model significantly outperforms other methods in terms of dynamics prediction.

## 5 Conclusion and Future Work

We present an object-oriented end-to-end neural network framework. This framework is able to learn object dynamics conditioned on both actions and object relations in an unsupervised manner. Its learned dynamics model exhibits strong generalization and interpretability. Our framework demonstrates that object perception and dynamics can be mutually learned and reveals a promising way to learn object-level knowledge by integrating both vision and interaction. We make one of the first steps in investigating how to design a self-supervised, end-to-end object-oriented dynamics learning framework that enables generalization and interpretability. Our learned dynamics model can be used with existing policy search or planning methods (e.g., MCTS and MPC). Although we use random exploration in the experiment, our model can integrate with smarter exploration strategies for better state sampling.

| Models | $\text{acc}_0$ | $\text{acc}_1$ | $\text{acc}_2$ | $\text{acc}_3$ |
|---|---|---|---|---|
| AC Model | 0.10 | 0.10 | 0.10 | 0.12 |
| CDNA | 0.46 | 0.54 | 0.62 | 0.75 |
| OODP | **0.70** | **0.70** | **0.78** | **0.92** |

Table 4: Accuracy of the dynamics prediction in unseen environments in Mars domain. $\text{acc}_n$ denotes $n$-error accuracy.

Our future work includes extending our framework for supporting long-term prediction, abrupt change prediction (e.g., object appearing and disappearing), and dynamic background (e.g., caused by a moving camera or multiple dynamic objects). As abrupt changes are often predictable from a long-term view or with memory, our model can incorporate memory networks (e.g., LSTM) to deal with such changes. In addition, the STN module in our model has the capability of learning disappearing, which is basically an affine transformation with zero scaling. For prediction with dynamic background (e.g., in FPS game and driving), we will incorporate a camera motion prediction network module similar to that introduced by Vijayanarasimhan et al. [45]. This module will learn a global transformation and apply it to the whole image to incorporate the dynamics caused by camera motion.

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
