[Supplementary Material]

# Supplementary Material for
# *Object-Oriented Dynamics Predictor*

**Guangxiang Zhu, Zhiao Huang, and Chongjie Zhang**
Institute for Interdisciplinary Information Sciences
Tsinghua University, Beijing, China
guangxiangzhu@outlook.com,hza14@mails.tsinghua.edu.cn,chongjie@tsinghua.edu.cn

## 1   Validation on greater maximum of object masks

OODP does not require the actual number of objects in an environment, but needs to set a maximum number. When they do not match, some learned object masks may be redundant, which does not affect the accuracy of predictions. We conduct an experiment to confirm this. As shown in Figure 1, when the maximum number is larger than the actual number, some learned object masks are redundant. Walls are detected by the two masks shown in the top right corner of the figure, while ladders are detected by the two masks in the bottom right corner. Furthermore, the accuracies of dynamics prediction are similar when the maximum number of object masks is different (0.840 and 0.842 for the maximum number 3 and 5, respectively).

Input frame

Object masks

Figure 1: Visualization of the masked images in unseen environments with the maximum number of object masks equal to 5.

## 2   Additional results of OODP

We supplement the learned masks (for Section 4.4 in the main body) in the unseen environments with novel object layouts and object appearance S2, which is shown in Figure 2. In addition, we further

Figure 2: Visualization of the learned masked images in S2 environments.

Figure 3: Cases to illustrate the performance of our model in novel environments including single (top) and multiple (bottom) dynamic objects.

provide more intuitive results of our performance on learned dynamics for perceptual understanding. Figure 3, 4 and 5 depict representative cases of our predictions in training and novel environments.

## 3    Architecture details

### 3.1    Background Extractor

As shown in Figure 6, the Background Extractor takes the form of a traditional encoder-decoder structure. The encoder alternates convolutions and ReLUs followed by two fully connected layers, while the decoder alternates deconvolutions and ReLUs. For all the convolutions and deconvolutions, the kernel size, stride, number of channels are 3, 2, and 64, respectively. The dimension of the hidden layer between the encoder and the decoder is 128. When OODP is trained in a large number of environments, convolutions with more channels (e.g., 128 or 256 channels) are needed to improve the expressiveness of Background Extractor. In addition, we replace the ReLU layer after the last deconvolution with tanh to output values ranged from -1 to 1.

### 3.2    CNNs in Object Detector and Dynamics Net

Figure 7 illustrates the architecture of the CNNs in Object Detector, which is similar with the encoder in [1]. Denote $Conv(F, K, S)$ as the convolutional layer with the number of filters $F$, kernel size $K$ and stride $S$. Let $R()$ and $BN()$ denote the ReLU layer and batch normalization layer [2]. The 5 con-

Figure 4: Two examples of predictions in training environments including one dynamic object.

Figure 5: Two examples of predictions in training environments including multiple dynamic and static objects.

$I^{(t)}$ → conv+ReLU → reshape → fully connected → deconv+ReLU $I^{(t)}_{bg}$

Figure 6: Architecture of Background Extractor.

volutional layers in the figure can be indicated as $R(BN(Conv(64, 5, 2)))$, $R(BN(Conv(64, 3, 2)))$, $R(BN(Conv(64, 3, 1)))$, $R(BN(Conv(32, 1, 1)))$, and $R(BN(Conv(1, 3, 1)))$, respectively.

The CNNs in Dynamics Net are connected in the order: $R(BN(Conv(16, 3, 2)))$, $R(BN(Conv(32, 3, 2)))$, $R(BN(Conv(64, 3, 2)))$, and $R(BN(Conv(128, 3, 2)))$. The last convolutional layer is reshaped and fully connected by the 128-dimensional hidden layer and the 2-dimensional output layer successively.

Figure 7: Architecture of the CNNs in Object Detector.

## 4    Implementation details

For OODP and all baseline models, the training images are collected by an agent with random policy exploring the environment and then the changed and changeless images are balanced via sampling. The images are down-sampled to size $80 \times 80 \times 3$. The parameters for training DDOP-p and OODP+p are listed as follows:

- In OODP-p, the weights for $\mathcal{L}_{\text{prediction}}$, $\mathcal{L}_{\text{entropy}}$, $\mathcal{L}_{\text{reconstruction}}$, $\mathcal{L}_{\text{consistency}}$, and $\mathcal{L}_{\text{background}}$ are 100, 0.1, 100, 1, and 1, respectively. While in OODP+p, we can remove the auxiliary losses ($\mathcal{L}_{\text{reconstruction}}$, $\mathcal{L}_{\text{consistency}}$, and $\mathcal{L}_{\text{background}}$) that have the similar functions to $\mathcal{L}_{\text{proposal}}$, and the weights for $\mathcal{L}_{\text{prediction}}$, $\mathcal{L}_{\text{entropy}}$, and $\mathcal{L}_{\text{proposal}}$ are 10, 1, and 1, respectively. In addition, all the $l_2$ losses are divided by $HW$ to keep invariance to the image size. To balance the positive and negative proposal regions, we use a weighted pixel-wise l2 loss for the proposal loss.

- Batch size is 16 and the maximum number of training steps is set to be $1 \times 10^6$. The size of the horizon window $w$ is 33.

- The optimizer is Adam [3] with learning rate $1 \times 10^{-4}$.