[Reviews · NeurIPS 2018]

Reviewer 1



This paper addresses the problem of action-conditional video prediction via a deep neural network whose architecture specifically aims to represent object positions, relationships, and interactions. The learned models are shown empirically to generalize to novel object configurations and to be robust to minor changes in object appearance. Technical Quality As far as I can tell the paper is technically sound. The experiments are well-designed to support the main claims. I especially appreciated the attempts to study whether the network is truly capturing object-based knowledge as a human might expect (rather than simply being a really fancy pixel -> pixel model). Clarity Honestly, this paper is a difficult read. A significant chunk of it is a description of a very complicated neural network architecture and I found it very hard to follow. That said, the architecture itself is one of the primary contributions and it must be described. Further, space limitations prevent a more relaxed walkthrough of the approach. Essentially, I'm afraid that I don't have any brilliant ideas about how to improve the reader's experience. I guess I would just say to the authors that it would be valuable for them to go over Section 3 with an eye toward giving high-level intuition and grounding the architectural choices in examples. If the authors can give the reader anything additional to help build this structure in their head and understand the intuitive function of each component, it will help. Originality/Significance The problem the paper is attempting to tackle is important and of broad interest. While certainly many of the components of the approach are familiar, the real trick is in composing them in a way that they will work together. The result is what seems to me to be a significant achievement and definitely a novel contribution. This approach can start with pixels, segment the scene into intuitive objects/object classes, and verifiably apply learned and generalizable object relations to predict movement. I'd say that's nothing short of remarkable; people are going to want to hear about this. I will say, however, that I do wish the paper had spent a little bit of time discussing the weakness/limitations of the approach. What are some types of dynamics that it is not likely to handle well and what are the core challenges? For instance, based on my understanding, it seems that it would not be well-suited to problems where objects can appear and/or disappear (since it only predicts movement for each object already on the screen). It would just be nice to look up and survey what this approach addresses and what significant hurdles remain. Overall This paper represents a significant advance in the challenge problem of action-conditional video prediction. Though I wish it were easier to extract high-level intuition and big ideas from the description of the network architecture, I understand the challenge of clearly presenting such a complicated structure. The approach is well-evaluated, experiments validating that the model is able to exploit object-relational structure as intended. ---After Author Response--- Having considered the other reviews and the author response I have decided to retain my original score. I frankly do not understand the arguments of the other reviewers, who seem to be dismissing the novel and important contribution of the paper (an end-to-end architecture that verifiably develops an object-oriented representation from raw perception) and demanding that the paper solve a bunch of other really hard, open problems (like dealing with variable numbers of objects, large numbers of object types, dynamic backgrounds, agent-centric positions, etc.). I continue to be impressed by the results and satisfied by the well-designed experiments that investigate the capabilities and behavior of the network. Though I frankly don't see where there's room, if the authors can squeeze the Mars Rover experiment into the paper as well, that would alleviate concern that this is a one-trick pony. To the authors, I suppose I can only say that the kind of discussion that is present in the response about what the model can and cannot do and how one might imagine extending/building upon these ideas to handle more complicated/interesting problems is precisely what the paper needs. If this paper is rejected, which would be a shame in my estimation, I strongly encourage the authors to submit this paper again. Perhaps a robust discussion along these lines in the paper itself would help to highlight the significance of the contribution in the context of these bigger goals.

Reviewer 2



Summary: This paper proposed an object-oriented dynamic predictor. This predictor can accurately predict dynamics and generalize to unseen states with the help of a class-specific detector. Strong points: + Object-oriented method provides a good representation for dynamic learning. This direction can be a very interesting one for future research. + The architecture is novel to combine static and dynamic objects. Weakness: - The experiments are only done on one game environment. More experiments are necessary. - This method seems not generalizable for other games e.g. FPS game. People can hardly do this on realistic scenes such as driving. Static Assumption too strong.

Reviewer 3



Update after rebuttal: After reading the rebuttal / other reviews, my conclusion is not changed significantly. This paper shows an end-to-end approach that works well for a specific set of MDPs (static background, dynamics objects) with <= 2 objects. Showing that the composition of modules can work in a simple setting is interesting. However, the paper can be compared with previous work along multiple angles, but each one feels unsatisfactory. - The dynamics aspects used are not comparable to previous work on learning dynamics (as authors cited, see e.g., https://arxiv.org/pdf/1612.00341.pdf that demonstrates their approach on 7 objects and that shows they can predict "the future" and interesting dynamical information e.g., mass). Also see https://arxiv.org/pdf/1705.10915.pdf, which predicts many frames into the future for *flexible* objects, and with high fidelity. Note that foreground/background setting is the same as here. This problem is actually much harder than learning object representations. To be convincing compared relative to this, the authors would have to evaluate their method on at least more than 1,2 objects. - As a work on unsupervised learning of object representations from sequential data, see e.g., https://www.di.ens.fr/willow/pdfscurrent/kwak2015.pdf (note that this doesn't use DNNs, but is easily combined with deep features). Hence, I stand by my previous conclusion: in the current state, the work would benefit greatly from - showing e.g., the merits of this method on a (challenging) RL task, as their introduction suggests as the motivation for this work. - or, comparing more challenging / richer dynamics aspects with previous work (see above) I do strongly encourage the authors to flesh out these aspects of their work. These would be additions that would definitely raise the delta significantly. --- Summary: The authors propose to learn environment models at the object-level rather than pixel-level (i.e., predict what the next state of a collection of pixels ("object") is). They train a neural network (OODP) end-to-end for this purpose. At the core is a Dynamics-net, which is action-conditioned; other information of the MDP is not used. Objects are divided into static and dynamic ones; the number of objects is fixed before training. A separate background splitter network is trained to predict all non-objects. An additional region proposal loss (OODP+p model) is used as well to focus dynamic object networks on "good" regions that likely contain a dynamic object. The authors also interpret the model by inspecting the scene parts predicted by the static / dynamic object models. Pro: - Using the structure of object-ness is a good idea to learn dynamics and potentially speeding up model-based methods. - Quantitative results of OODP suggest the method works well (lower state-prediction "accuracy" than other methods). - Authors evaluate on k-to-m generalization (train on k envs, test on m envs) for different backgrounds and layouts. Con: - The authors only evaluate on data from a single MDP, which seems quite limited. Does the performance gains appear for other games as well? - How do we choose n_O beforehand? - How does the model deal with a variable number of objects at test-time? What if objects disappear (e.g., an enemy is killed)? How would we extend the proposed architecture? It would be nice to include a discussion of this aspect. - The authors do not evaluate on any real RL task (i.e., with exploration) / integrate OODP with model-based RL approaches; the only MDP information seems to come from the action-conditioning and the fixed sequence of states for training. The contribution of this paper thus seems to mostly be in the supervised learning domain (although the motivation mentions RL). Therefore, I'm not so sure that L289 is warranted -- "We make one of the first steps in investigating whether object-oriented modeling is beneficial for generalization and interpretability in ***deep reinforcement learning***". Clarity: - What are the units of "accuracy"? Reproducibility: - ODDP is a complicated model with many moving parts and a complicated total loss function (involves 6 losses!). It thus seems hard to fully replicate the authors' results. It would thus be very beneficial to release code and trained models.